# Nanopore sequencing of SARS-CoV-2: Comparison of short and long PCR-tiling amplicon protocols

**Broňa Brejová**[1©]*, **Kristína Boršová**[2,3©], **Viktória Hodorová**[4], **Viktória Čabanová**[2], **Askar Gafurov**[1], **Dominika Fričová**[5], **Martina Neboháčová**[4], **Tomáš Vinař**[6], **Boris Klempa**[2], **Jozef Nosek**[4]*

1 Department of Computer Science, Faculty of Mathematics, Physics and Informatics, Comenius University in Bratislava, Bratislava, Slovak Republic, 2 Institute of Virology, Biomedical Research Center of the Slovak Academy of Sciences, Bratislava, Slovak Republic, 3 Department of Microbiology and Virology, Faculty of Natural Sciences, Comenius University in Bratislava, Bratislava, Slovak Republic, 4 Department of Biochemistry, Faculty of Natural Sciences, Comenius University in Bratislava, Bratislava, Slovak Republic, 5 Institute of Neuroimmunology, Slovak Academy of Sciences, Bratislava, Slovak Republic, 6 Department of Applied Informatics, Faculty of Mathematics, Physics and Informatics, Comenius University in Bratislava, Bratislava, Slovak Republic

© These authors contributed equally to this work.
* brejova@dcs.fmph.uniba.sk (BB); jozef.nosek@uniba.sk (JN)

**Data Availability Statement:** https://www.ebi.ac.uk/ena/browser/view/PRJEB44303.

**Funding:** The research was supported by grants from the Slovak Research and Development Agency (https://www.apvv.sk; APVV-18-0239 to

## Abstract

Surveillance of the SARS-CoV-2 variants including the quickly spreading mutants by rapid and near real-time sequencing of the viral genome provides an important tool for effective health policy decision making in the ongoing COVID-19 pandemic. Here we evaluated PCR-tiling of short (~400-bp) and long (~2 and ~2.5-kb) amplicons combined with nanopore sequencing on a MinION device for analysis of the SARS-CoV-2 genome sequences. Analysis of several sequencing runs demonstrated that using the long amplicon schemes outperforms the original protocol based on the 400-bp amplicons. It also illustrated common artefacts and problems associated with PCR-tiling approach, such as uneven genome coverage, variable fraction of discarded sequencing reads, including human and bacterial contamination, as well as the presence of reads derived from the viral sub-genomic RNAs.

## Introduction

Massive spreading of severe acute respiratory syndrome coronavirus 2 (SARS-CoV-2) within the human population began in December 2019 in Wuhan, Hubei Province, China [1–3]. In the following weeks, the virus has been quickly transmitted all over the globe. As of June 22, 2021, it infected more than 178 million humans and caused over 3.9 million deaths (https://arcg.is/0fHmTX; [4]). The 29,903-nt long genomic RNA sequence of the SARS-CoV-2 strain Wuhan-Hu-1 (Genbank/RefSeq acc.nos. MN908947 / NC_045512; [1]) and related isolates [2, 3] were determined early in 2020 and facilitated rapid development of molecular diagnostics as well as the analysis of additional isolates from other geographical regions of the world. More than 2 million SARS-CoV-2 genome sequences are available in the GISAID repository (http://

JN, PP-COVID-20-0017 to BK), the Scientific Grant Agency (https://www.minedu.sk/vedecka-grantova-agentura-msvvas-sr-a-sav-vega/; VEGA 1/0463/20 to BB, VEGA 1/0458/18 to TV, VEGA 1/0027/19 to JN, VEGA 1/0136/20 to MN), and the European Union's Horizon 2020 research and innovation program (https://ec.europa.eu/programmes/horizon2020/; EVA-GLOBAL project #871029 to BK and PANGAIA project #872539 to TV). The research was also supported in part by the Operation Program of Integrated Infrastructure (OPII) projects ITMS2014: 313011ATL7 and ITMS2014+: 313021X329 (Advancing University Capacity and Competence in Research, Development and Innovation), co-financed by the European Regional Development Fund (https://ec.europa.eu/regional_policy/en/funding/erdf/). The funders had no role in study design, data collection and analysis, decision to publish, or preparation of the manuscript.

**Competing interests:** The authors have declared that no competing interests exist.

www.gisaid.org, June 22, 2021), thus representing an unprecedented resource for the scientific community and public health officials.

Rapid, cost-effective, and near real-time genome sequencing of the SARS-CoV-2 variants combined with epidemiological data provides an important resource not only for understanding the virus transmission, its genetic alterations and evolution, but also for making the policy decisions in combating the pandemic [5]. Monitoring sequence diversification plays an essential role in continual refinement of molecular diagnostics (*e.g.*, redesigning the primers for nucleic acid amplification techniques [6] or development of screening tools for variants of concerns (VoC) and those evading the immune response [7, 8]). This underscores the importance of genomic epidemiology, although the elucidation of direct links between particular mutation (s) and the virus spreading or clinical implications still represents a challenging task [9–19].

The SARS-CoV-2 sequences were determined using a range of experimental approaches based on metagenomics, sequence capture or enrichment, amplicon pools by deploying short (*e.g.*, Illumina) or long-read (*e.g.*, Pacific Biosciences, Oxford Nanopore Technologies) sequencing platforms. Of these, nanopore sequencing becomes increasingly popular as in addition to sequencing of viral genomic RNA it also permits transcriptome mapping, characterization of sub-genomic RNA molecules, and identification of modified nucleotides in the viral genome [20–22].

The protocol for nanopore sequencing of tiled PCR-generated amplicon pools has been developed by the Artic Network (https://artic.network/) for sequencing of Ebola, Zika, and Chikungunya genomes [23, 24]. In January 2020, the original protocol was promptly adjusted for rapid sequence determination of SARS-CoV-2 RNA prepared directly from clinical samples such as nasopharyngeal or oropharyngeal swabs. Additional studies described its modifications including alternative primer schemes and different amplicon sizes or different sequencing chemistries [25–36]. Its further improvements resulted in simplification of the sequencing library preparation, shortened hands-on time, and increased sample multiplexing (up to 96) that decreased the reagent costs to about £10 per sample, making this approach affordable for epidemiologic surveillance of the pandemic [36]. Importantly, rigorous comparison of nanopore sequencing with Illumina short reads technology demonstrated that in spite of relatively high error rates in individual nanopore reads, highly accurate consensus single nucleotide variant (SNV) calling with >99% sensitivity and >99% precision can be achieved with a minimum of about 60-fold coverage [37].

In this study, we compare the performance of several PCR-tiling based protocols which were evaluated as part of our efforts to sequence isolates of SARS-CoV-2 from Slovakia collected between March 2020 and March 2021. Using the generated sequence data, we investigate the nature of common problems and artefacts associated with this approach. We compare the sequencing results obtained from the libraries containing multiplexed barcoded SARS-CoV-2 samples made of ~400-bp, ~2-kb, and ~2.5-kb long overlapping amplicon pools as well as the combination of short and long amplicons. Our results show that sequencing of long amplicons clearly outperforms the original protocol based on shorter amplicons in terms of lower coverage variation and overall quality of the final sequence consensus. We also compare the performance of MinION runs with the standard (FLO-MIN106) and Flongle (FLO-FLG001) flow cells differing by nominal pore counts, *i.e.* 2048 (split into four sets of 512 each) and 126, respectively.

## Results and discussion

The PCR-tiling amplification combined with nanopore sequencing was employed for genome sequence analysis of 152 SARS-CoV-2 isolates from Slovakia (**S1 Table**). The genome

**Table 1. Overview of the MinION sequencing runs.**

| Batch | Amplicons | Barcodes used | Flow cell type | Flow cell QC[1] | Run time | Yield (Gbp) | Experiment date |
|---|---|---|---|---|---|---|---|
| UKBA-2 | 400 bp | 11 | FLO-MIN106 | 464[2] | 20 h 15 min | 0.90 | 2020-07-24 |
| | | | FLO-FLG001 | 18 | 20 h 44 min | 0.11 | 2020-07-24 |
| | | | FLO-FLG001 | 56 | 20 h 22 min | 0.38 | 2020-07-24 |
| | 2 kb | 12 | FLO-MIN106 | 1583 | 4 h 3 min | 2.16 | 2020-07-28 |
| | | | FLO-FLG001 | 63 | 37 h 33 min | 0.83 | 2020-07-28 |
| | | | FLO-FLG001 | 42 | 24 h 50 min | 0.47 | 2020-07-28 |
| UKBA-3 | 400 bp | 10 | FLO-MIN106 | 1126 | 4 h 4 min | 0.96 | 2020-09-30 |
| | 2 kb | 10 | FLO-MIN106 | 1267[2] | 4 h 55 min | 2.17 | 2020-09-30 |
| | 400 bp + 2 kb | 10 | FLO-MIN106 | 374[2] | 4 h 57 min | 0.75 | 2020-09-30 |
| UKBA-4 | 2 kb | 12 | FLO-MIN106 | 673[2] | 5 h 39 min | 1.88 | 2020-12-10 |
| | | | FLO-FLG001 | 32 | 23 h 30 min | 0.39 | 2020-12-10 |
| | | | FLO-FLG001 | 67 | 23 h 21 min | 0.60 | 2020-12-10 |
| UKBA-6 | 2 kb | 11 | FLO-MIN106 | 696[2] | 3 h 12 min | 1.11 | 2021-01-07 |
| UKBA-10 | 2 kb | 24 | FLO-MIN106 | 1031 | 4 h 28 min | 2.02 | 2021-01-29 |
| UKBA-11 | 2 kb | 24 | FLO-MIN106 | 1042[2] | 4 h 33 min | 2.10 | 2021-02-03 |
| UKBA-12 | 2 kb | 23 | FLO-MIN106 | 808[2] | 2 h 50 min | 1.21 | 2021-02-05 |
| UKBA-19 | 2.5 kb orig. | 24 | FLO-MIN106 | 667[2] | 18 h 54 min | 4.98 | 2021-03-16 |
| UKBA-21 | 2.5 kb mod. | 12 | FLO-MIN106 | 824[2] | 1 h 54 min | 0.51 | 2021-03-24 |

[1]—the number of active pores at the start of a sequencing run.

[2]—these flow cells were re-used after washing with the buffer containing nuclease.

sequences were obtained using primer schemes generating either ~400-bp (Artic Network version V3, https://github.com/artic-network/artic-ncov2019), ~2-kb [35], or ~2.5-kb long amplicons [27].

To compare primer sets for short and long amplicons and/or flow cell types, three different batches (UKBA-2, UKBA-3 and UKBA-4 in **Table 1**) consisting of 10–12 multiplexed samples were sequenced using multiple strategies. In batches UKBA-2 and UKBA-3, the same biological material was amplified using the primer schemes for both 400-bp and 2-kb long amplicons. Moreover, in batches UKBA-2 (400-bp and 2-kb long amplicons) and UKBA-4 (2-kb long amplicons), we loaded the same sequencing libraries to both the standard and Flongle flow cells. **Fig 1** shows the comparison of the fraction of samples in a sequencing run successfully sequenced at various cut-offs measuring the total amount of sequencing normalized by the number of samples in the run. In both batches UKBA-2 and UKBA-3, 2-kb amplicons clearly outperform 400-bp amplicons. Sequencing of a mixture of longer and shorter amplicon pools provided comparable results to sequencing longer amplicons alone, perhaps because the mixture was enriched in the long amplicons. Finally, the Flongle and standard flow cells are similarly successful at comparable sequencing volumes. However, there are two disadvantages to using the Flongle flow cells. First, the Flongle cannot be washed and reused, its entire capacity is used for a single experiment. Second, since there is a large variance in the amount of data produced by a single Flongle flow cell (in our experiments, the number of active pores in Flongles ranged between 18 and 67 pores and produced between 110 and 830-Mbp—see **Table 1**), the capacity may be insufficient to completely recover sequences of 10 or more multiplexed samples. We consider as an important advantage that the runs using the standard flow cells can be terminated when sufficient data is collected, and thus these flow cells can be reused in further experiments after washing with the buffer containing nuclease (*i.e.*, EXP-WSH003 or

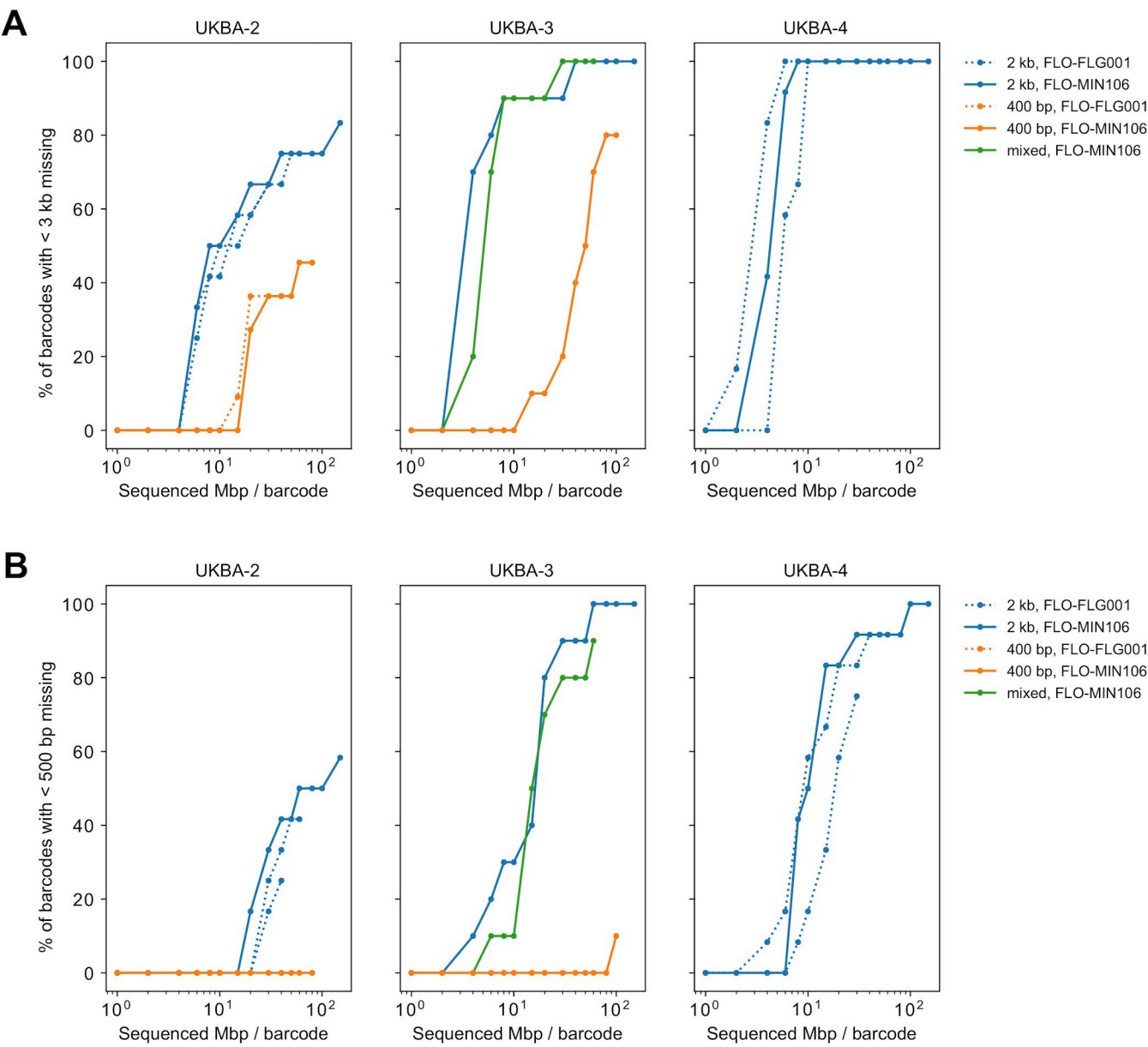

**Fig 1. The percentage of successfully sequenced multiplexed samples over time.** A sample is considered as successfully sequenced if the resulting sequence produced by the Artic pipeline has fewer than 500-bp (A) or 3-kb (B) marked as missing bases. Each run is represented by several time points, each point showing the percentage of successfully sequenced barcodes (y-axis) upon reaching a specified amount of sequenced data per barcode (x-axis).

EXP-WSH004). Moreover, the standard flow cells allow simultaneous sequencing of a greater number of barcoded samples with a longer run.

Note that batch UKBA-2 included samples with low product concentrations after PCR amplification. As a result, three samples (barcodes 02, 06 and 11) could not be completed reliably even after combining data from all six sequencing runs. Batches UKBA-3 and UKBA-4 contained only samples with Cq values from RT-qPCR below 26. **S1 Fig** shows the amount of missing sequence in individual samples plotted against possible explanatory variables, namely the Cq values, amplicon concentration, and RNA sample storage time prior to amplification. Although the expected trends are in some cases observable, they are not followed universally.

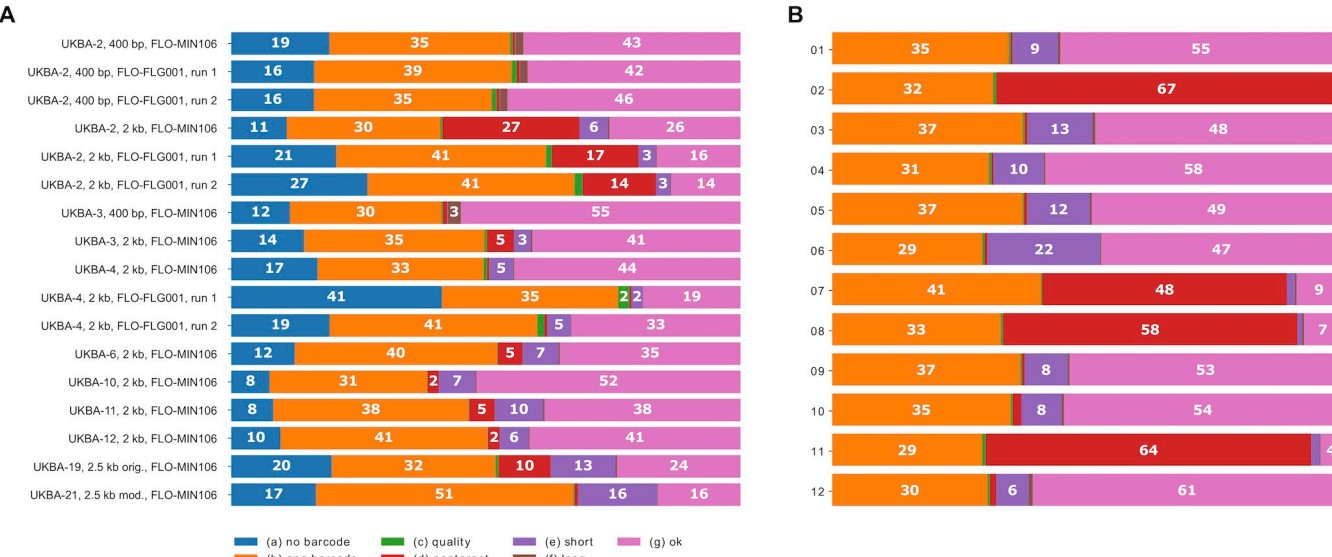

**Fig 2. Reasons for discarding reads in the Artic pipeline.** The sequencing reads must pass through a series of filters to ensure correct sample assignment and the read quality. The bar graphs show the percentage of reads discarded for various reasons as well as those passing all filters. Panel (A): Summary per run. Panel (B): Detailed per-barcode analysis for UKBA-2 samples, 2-kb amplicons, standard flow cell. Group (a): reads without barcode identification. Group (b): reads with only one barcode (Artic pipeline requires barcodes on both ends to ensure that the whole read was sequenced and to decrease the probability of barcode bleeding). Group (c): low-quality reads (base caller quality less than 7). Group (d): reads that do not align to the SARS-CoV-2 reference. Group (e): reads that are too short (likely due to fragmentation). Group (f): reads that are too long (*i.e.* chimeric reads). The pipeline keeps reads of lengths between 1500 and 3000 for 2-kb amplicons, between 350 and 619 for 400-bp amplicons. The reads passing all filters are included in group (g).

Using the Artic pipeline for further analysis, sequencing reads must first pass a series of filters to ensure no barcode bleeding and to remove possible contamination. The number of reads passing these filters and used for the identification of variants in the final step of the pipeline varied between runs. In our experiments their fraction comprises between 14 and 55% (**Fig 2A**). Majority of failed reads (41–78% of all reads) are due to the low quality or incompleteness, often leading to inability to recognize one or both barcodes (groups (a)-(c)). While there are no clear differences between short and long amplicon protocols, with 2-kb amplicons these low-quality reads seem to be more prevalent on the Flongle runs compared to the standard flow cells.

Interestingly, in some runs, up to 27% of reads that pass the base quality filters do not map to the target reference genome. In particular, four samples in batch UKBA-2 of 2-kb amplicon run (barcodes 02, 07, 08 and 11) have a very high fraction of non-target reads (**Fig 2B**). The majority (82–96%) of these reads map to the human genome, and a smaller fraction (0.3–9%) map to bacterial genomes, including the species colonizing human oral cavity and respiratory tract (*e.g.*, *Actinomyces graevenitzii*, *Haemophilus parainfluenzae*, *Leptotrichia* spp., *Prevotella* spp., *Pseudomonas aeruginosa*, *Rothia mucilaginosa*, *Streptococcus pneumoniae*, *S. mitis*, *S. parasanguinis*, *S. salivarius*, *Tannerella forsythia*, *Veillonella parvula*). All four samples showed a lower viral load (*i.e.*, Cq value > 30) in RT-qPCR assays, and the amplification in the PCR-tiling protocol resulted in lower product yield. Human and bacterial reads represent artefacts apparently resulting from a non-specific amplification of contaminating nucleic acids present in clinical samples.

We have also observed that some amplicons originate from sub-genomic RNAs that co-purify with the SARS-CoV-2 genomic RNA. It has been demonstrated that the amount of sub-genomic RNAs correlates with the disease severity. As these molecules are strongly repressed in asymptomatic patients [38], their proportion in the sequencing data can serve as a

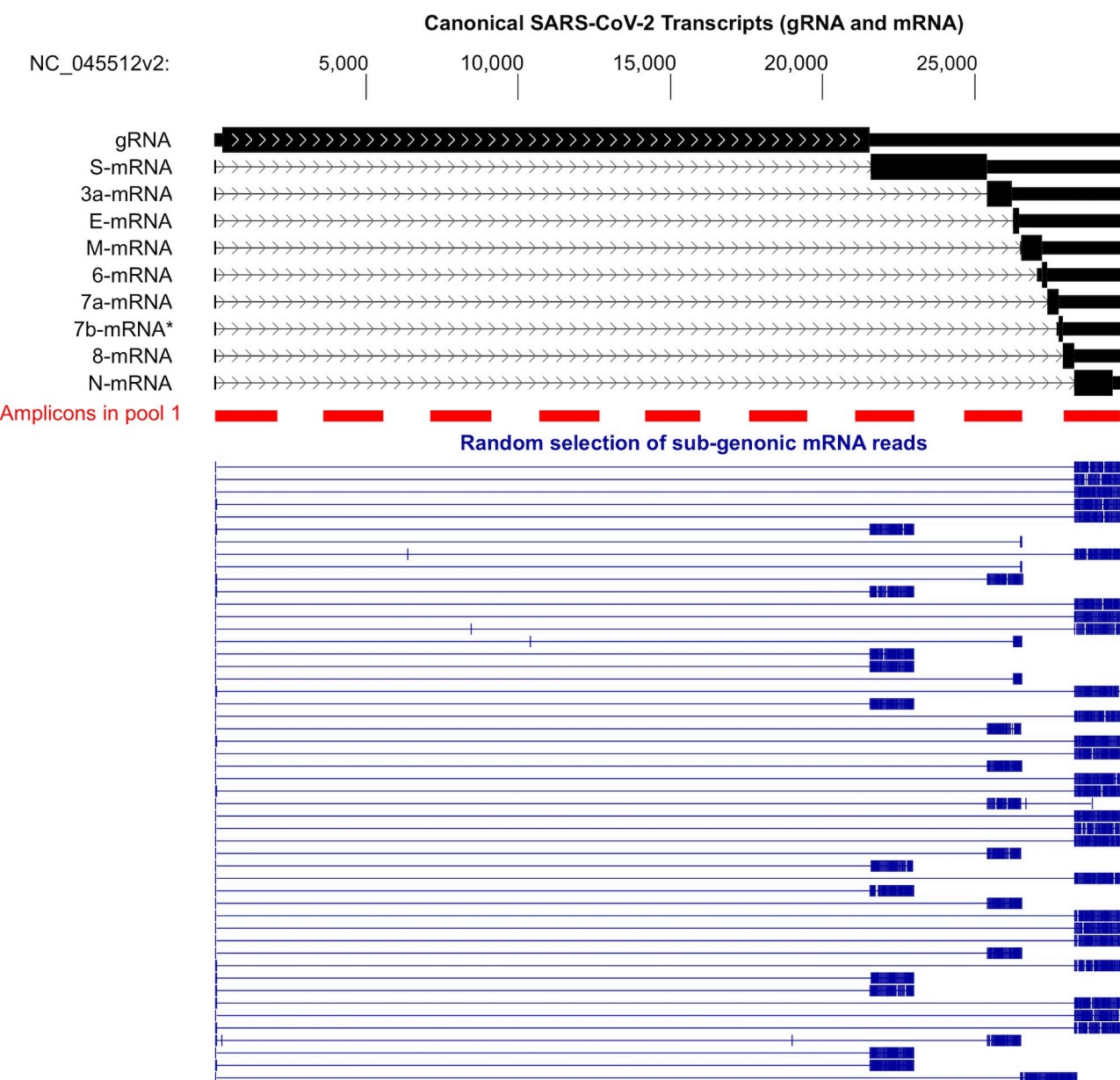

**Fig 3. Reads derived from the sub-genomic RNAs.** Sub-genomic RNAs (black), amplicons of primer pool 1 from the 2-kb primer set (red), and spliced alignments of a random sample of 50 reads from barcode 07 from UKBA-2 run with 2-kb amplicons classified as sub-genomic (blue). Visualization was created by the UCSC genome browser [40].

molecular marker. The most abundant reads are derived from the N mRNA [39]. The sub-genomic RNAs are generated in the process of the virus replication/transcription [5] and start with a leader sequence originating from the untranslated 5' end of the viral genome, followed by a downstream sequence containing a particular open reading frame. The leftmost primer in both 400-bp and 2-kb primer sets investigated in this study is contained within the leader sequence. This facilitates amplification of sub-genomic RNAs with appropriate right primers (**Fig 3**). **Table 2** lists the fraction of selected sub-genomic RNAs among reads that could be aligned to the SARS-CoV-2 genome. These fractions are relatively low, with the remaining

**Table 2. Percentage of sub-genomic RNAs out of reads that align to the SARS-CoV-2 genome and can be demultiplexed were considered.**

| batch | amplicon size | ORF3a | Gene E | Gene M | Gene N | Gene S | genome |
|---|---|---|---|---|---|---|---|
| UKBA-2 | 2 kb | 2.4 | 2.5 | 1.4 | 3.5 | 0.8 | 89.3 |
| | 400 bp | 0.0 | 0.0 | 0.1 | 0.5 | 0.0 | 99.3 |
| UKBA-3 | 2 kb | 1.0 | 1.2 | 1.1 | 1.5 | 0.6 | 94.6 |
| | 400 bp | 0.0 | 0.0 | 0.1 | 0.2 | 0.0 | 99.7 |
| UKBA-4 | 2 kb | 0.9 | 0.9 | 1.0 | 1.0 | 0.8 | 95.3 |
| batch | barcode # | ORF3a | Gene E | Gene M | Gene N | Gene S | genome |
| UKBA-2 (2 kb) | 01 | 0.9 | 1.8 | 0.9 | 1.5 | 0.5 | 94.4 |
| | 02 | 0.3 | 0.3 | 1.0 | 0.4 | 0.1 | 97.8 |
| | 03 | 2.4 | 2.7 | 1.8 | 3.7 | 1.5 | 87.9 |
| | 04 | 2.8 | 2.6 | 1.2 | 2.2 | 1.0 | 90.3 |
| | 05 | 2.0 | 2.5 | 2.0 | 2.1 | 0.8 | 90.6 |
| | 06 | 4.5 | 5.0 | 2.0 | 5.4 | 0.0 | 83.1 |
| | 07 | 4.0 | 0.5 | 0.9 | 14.3 | 3.0 | 77.3 |
| | 08 | 3.2 | 3.4 | 0.1 | 0.0 | 0.0 | 93.2 |
| | 09 | 1.3 | 2.2 | 1.7 | 2.5 | 1.3 | 91.0 |
| | 10 | 0.9 | 1.8 | 1.2 | 2.2 | 0.3 | 93.7 |
| | 11 | 7.5 | 0.0 | 0.9 | 3.5 | 0.0 | 87.9 |
| | 12 | 2.1 | 1.5 | 0.5 | 4.0 | 1.3 | 90.5 |

Only genes with the highest numbers of sub-genomic RNA reads are shown. Top: statistics for different MinION runs with the standard flow cells. Bottom: statistics for different barcodes of batch UKBA-2, 2-kb amplicons.

sub-genomic RNAs being even more rare. However, the fractions vary among the samples. In UKBA-2 run with 2-kb amplicons, the highest fraction of 14.3% was observed for the gene N mRNA in barcode 07 and the fraction of 7.5% was observed for the ORF3a mRNA in barcode 11. Some of these sub-genomic amplicons are discarded from the analysis as too short, while others lead to uneven coverage in the amplicon regions containing gene starts (**Fig 4**).

From these pilot experiments, we conclude that even though 400-bp amplicons have a lower percentage of discarded reads (**Fig 2**), they produce fewer finished sequences at a comparable overall amount of sequence data (**Fig 1**). The reason is a very uneven coverage of individual amplicons (**Fig 4**). This is observed in both sets of primers, but for the 400-bp amplicons we see a much lower coverage in the worst covered regions (**Fig 5**). Additional sequencing runs (UKBA-6, UKBA-10, UKBA-11, and UKBA-12) were performed with long 2-kb amplicons on standard MinION flow cells with similar results (**Fig 2A**; **S2 Fig**).

To investigate if a different primer scheme for generating long amplicons can solve the problem with uneven coverage (in particular, see amplicon 13 in **Fig 4** which partially covers the S gene region important for identification of the SARS-CoV-2 Variants of Concerns), we also tested the 2.5-kb primer panel [27]. Except for the leftmost primer, the primer positions in this panel differ from those of the 2-kb scheme. We have performed two sequencing runs with the 2.5-kb primer set (UKBA-19, UKBA-21). In the first experiment, we have noticed an almost complete drop of coverage in the last amplicon derived from the 3' end of the genome; for the second experiment, we have replaced the primers for the right-most amplicon with the right-most primer pair from the 2-kb panel, which mitigated the issue. Comparing the coverage of individual amplicons between the 2-kb and 2.5-kb schemes (**Fig 6**), the coverage in the 2.5-kb scheme indeed appears to be more even. **Fig 5** illustrates that our modification of the 2.5-kb scheme leads to a particularly small difference between the median coverage and coverage of the lowest 10% of the genome, which may result in fewer regions with insufficient

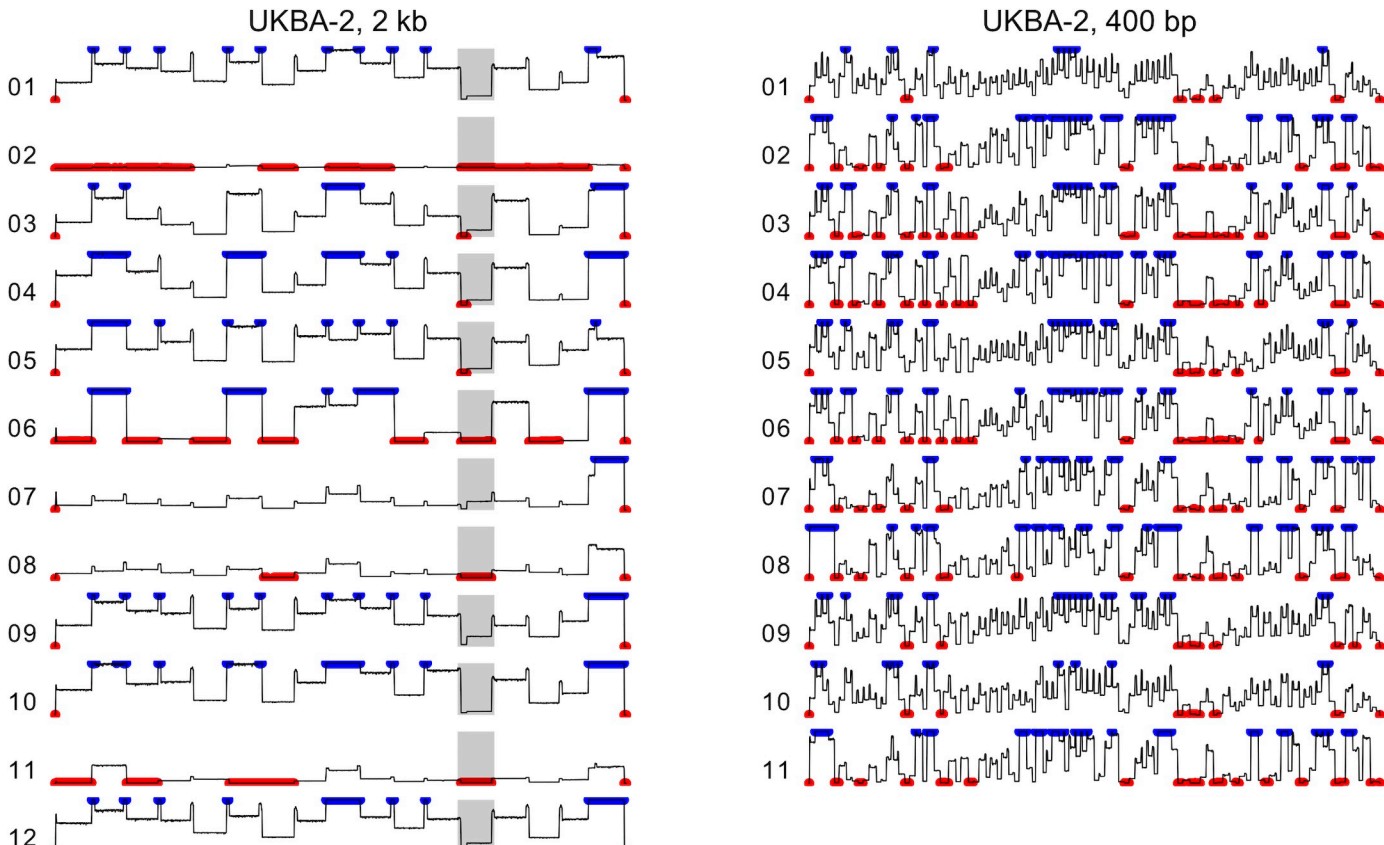

**Fig 4. Coverage along the genome in two MinION runs for batch UKBA-2.** In both runs, an initial portion of the run containing on average 40-Mbp of sequencing data per barcode was used. Coverage values higher than 1000 were clipped at this value and are shown in blue. Coverage below 20 (default Artic cutoff) is shown in red. Medians of 10-bp windows are shown for smoothing. The very starts and ends of the genome are not covered by amplicons and are thus displayed in red. Shaded area in the left column corresponds to amplicon 13. Some barcodes have a visible dip in the coverage at the left end of this amplicon; this difference in coverage is caused by reads originating from sub-genomic RNAs corresponding to the gene S. Similar plots for additional runs are shown in **S2 Fig**.

coverage. However, we have also noticed a higher percentage of failed reads, with only 24% (UKBA-19) and 16% (UKBA-21) reads passing all filters and being usable for variant identification (**Fig 2A**). Further analysis revealed a notable increase in single-barcode reads (group (b)) and shorter than expected reads (group (e)), pointing to difficulties in amplifying and sequencing longer fragments. More experiments are required to determine whether the 2.5-kb scheme results in more fully-assembled genomes over the 2-kb scheme.

## Conclusions

In this paper, we have compared three versions of PCR-tiling protocol for sequencing SARS-CoV-2 genomes from clinical samples on the MinION platform. Our results have shown that even though the protocol based on short 400-bp amplicons generally produces more usable data, the coverage of individual amplicons varies widely which may result in difficulties in recovering individual mutations in under-represented amplicons. Uneven genome coverage has been reported elsewhere [28, 31] and occurs also in the data produced by other research groups (**S3 Fig**), but it can be reduced by the protocol optimization [31, 36]. In comparison, longer amplicons tend to produce close-to-finished genomes more quickly, generally requiring smaller amounts of raw data produced per barcode sequenced. However, protocols based on

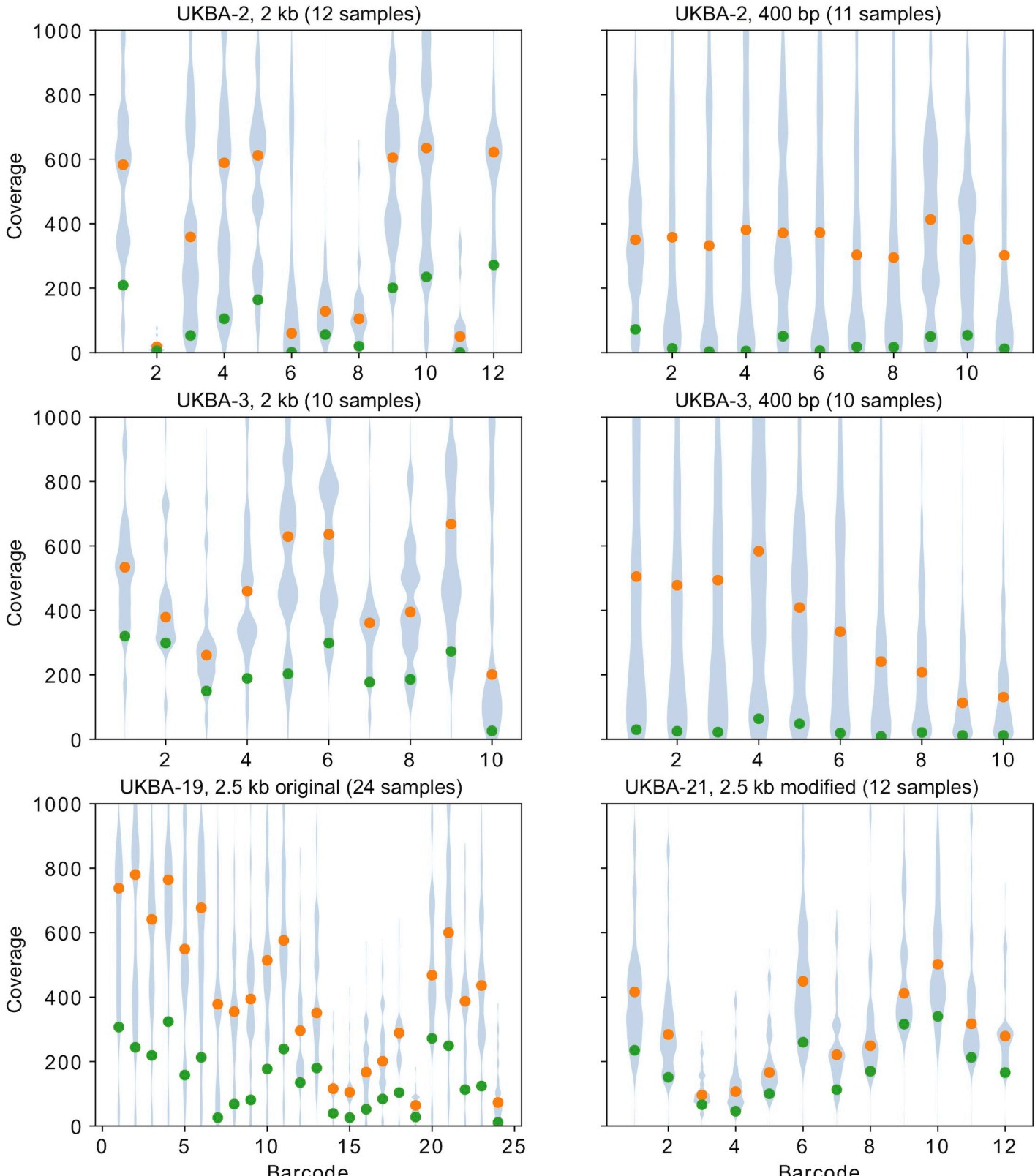

**Fig 5. Coverage distribution in different sequencing runs.** For each barcode, coverage by reads passing the Artic filter was computed along the genome (shown in **Fig 4** and **S2 Fig**) and the distribution of the coverage values was summarized as a violin plot (blue), cropped at coverage 1000. Orange dots represent median coverage

and green dots 10th percentile (approx. 3,000 bases of the genome have coverage below the green dot value). In all runs, an initial portion containing on average 40-Mbp of sequencing data per barcode was used.

long amplicons produce a higher percentage of reads that are unsuitable for further analysis with the Artic pipeline, likely due to a combination of fragmentation of synthesized molecules and prematurely aborted molecules during sequencing. The longer amplicon protocols are also less suitable for applications, where original RNA molecules in clinical samples may already be fragmented. Generally, the Flongle flow cells performed worse in sequencing multiplexed libraries containing barcoded samples than regular MinION flow cells, which have an added advantage of ability to adjust the length of the run based on the library and individual sample quality.

Interestingly, PCR-tiling protocols were able to also pick up sub-genomic RNA transcripts, and the proportion of these transcripts varied between samples. Since increased levels of sub-genomic transcripts are correlated with severe cases of COVID-19, these protocols could be optimized to detect the levels of sub-genomic transcripts more accurately and used as a biomarker for disease severity.

In our experiments, the divergence of samples from the SARS-CoV-2 reference sequence ranged from 0.02% to 0.13%, with higher divergence in case of newer samples. We did not observe these differences introducing problems in bioinformatic analysis, as tools used to

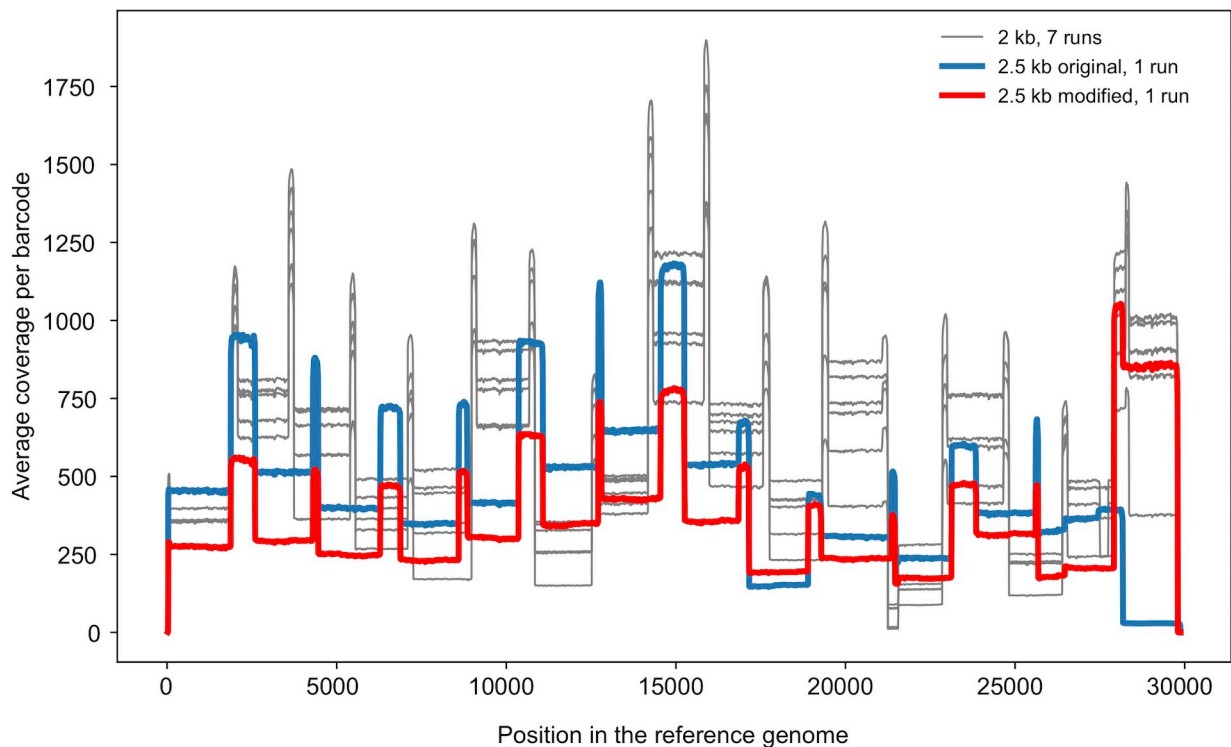

**Fig 6. Genome coverage by long amplicons.** Average coverage along the genome for seven runs with 2-kb amplicons (batches UKBA-2,3,4,6,10,11,12) and two runs with 2.5-kb amplicons (UKBA-19 with the original primer set and UKBA-21 with the last primer pair replaced by its counterpart from the 2-kb scheme). Each line depicts the average coverage over all samples in a run at the time point when 40-Mbp per sample was sequenced on average. Medians of 50-bp windows are shown for smoothing. Note a drop-out in the amplicon 13 (2-kb scheme) which covers a 3' end of orf1b and about a third of the S gene including the region associated with mutations in Variants of Concern such as B.1.1.7.

analyze sequencing reads in this study were designed to perform consistently across a broad range of sequence divergence.

Mutations at sites overlapping PCR primers, however, can decrease the efficiency, or even completely disable amplification of some regions, which can be detected by examining neighbouring amplicons overlapping the position of the primer. Thus, some primers may need to be modified as new mutations develop in the virus population. Readjusting the primer pools has also been reported as a strategy helping to increase the efficiency of amplification in poorly covered regions [29, 31]. Regardless of the reason, the primer readjustment is a much easier task for long amplicon protocols, since one has to consider much smaller primer sets (218 primers for 400-bp protocols vs. 28 primers for 2.5-kb protocol).

It is evident that effective epidemiologic surveillance of the pandemic is strongly dependent on systematic sequencing of SARS-CoV-2 isolates. The combination of PCR-tiling of overlapping amplicon pools with nanopore sequencing on the MinION platform from Oxford Nanopore Technologies is one of the most powerful and versatile means for acquisition of viral sequences. Yet, as demonstrated in this study, the pros and cons of a particular protocol must be taken into account to ensure that the sequencing results will be of the highest quality, which is an essential prerequisite for their utility in fighting the pandemic.

As of August 2021, long amplicon protocols are routinely used in our genomic surveillance pipeline in Slovakia to sequence as many as 96 barcoded samples in a single run. Both systematic comparison of 2-kb and 2.5-kb long amplicon protocols on sequencing runs with large numbers of samples as well as further optimization of primer pools are important issues for further study towards improvement of the SARS-CoV-2 sequencing efficiency.

## Materials and methods

### Collection of samples and RNA preparation

Oropharyngeal swabs of patients with suspected COVID-19, collected between March 30, 2020 and March 19, 2021, were preserved in 2–3 ml of viral transport medium and delivered to the laboratory of the Biomedical Research Centre of the Slovak Academy of Sciences in Bratislava, Slovakia in frame of the routine RT-qPCR diagnostics for SARS-CoV-2. Initially (UKBA-2 samples), 100 μl of the swab medium was used for the RNA extraction using the Zymo Research Quick-RNA™ Viral 96 Kit (Zymo Research, Irvin, California, USA). Resulting RNA was eluted to 20 μl of nuclease free water. For all other specimens, the Biomek i5 Automated Workstation (Beckman Coulter, Indianapolis, Indiana, USA) was employed using the RNAdvanced Viral kit (Beckman Coulter, Indianapolis, Indiana, USA). In this case, RNA was extracted from 200 μl of swab medium and eluted to 40 μl of nuclease free water.

### Real-time quantitative PCR (RT-qPCR)

In frame of the routine RT-qPCR diagnostics, presence of SARS-CoV-2 RNA was detected by vDETECT COVID-19 RT-qPCR kit, rTEST COVID-19 RT-qPCR kit or rTEST COVID-19 RT-qPCR ALLPLEX kit (MultiplexDX, Bratislava, Slovakia) targeting RNA-dependent RNA polymerase (RdRp) and Envelope (E) genes. The RT-qPCR assays were carried on QuantStudio™ 5 Real-Time PCR System (Applied Biosystem, Foster City, California, USA).

### Library preparation and DNA sequencing

The sequencing libraries were constructed using a ligation kit (SQK-LSK109) essentially as described in a PCR-tiling of COVID-19 virus protocol (PTC_9096_v109_revF_06Feb2020; Oxford Nanopore Technologies, Oxford, UK) with minor modifications. Briefly, RNA samples

extracted from swabs positive for the presence of SARS-CoV-2 in RT-qPCR assay (quantification cycle (Cq) values 13.46–32.03; **S1 Table**) were converted into cDNA using a SuperScript IV reverse transcriptase (Thermo Fisher Scientific, Waltham, Massachusetts, USA) or LunaScript® RT SuperMix Kit (New England Biolabs, Ipswich, Massachusetts, USA). For each sample, the overlapping amplicons were generated using a Q5® Hot Start High-Fidelity DNA polymerase (New England Biolabs, Ipswich, Massachusetts, USA) and the primer pools spanning the SARS-CoV-2 genome sequence (*i.e.*, 400-bp Artic nCoV-2019 V3 panel (https://github.com/artic-network/artic-ncov2019) purchased from Integrated DNA Technologies (IDT, Coralville, Iowa, USA, cat.no. 10006788) and the 2-kb [35] and 2.5-kb schemes [27], custom synthesized by Microsynth AG, Balgach, Switzerland). The same cycling program was used for all amplicon types (*i.e.*, 30 sec initial denaturation at 98˚C, followed by 25 to 35 cycles of 15 sec at 98˚C (denaturation) and 5 min at 65˚C (combined annealing and polymerization), and cooling to 4˚C). The amplifications were performed in two separate reactions and the overlapping amplicons were pooled, purified using an equal volume of AMPure XP magnetic beads (Beckman Coulter, Brea, California, USA) and quantified using a Qubit 3.0 spectrophotometer and dsDNA Broad Range Assay Kit (Thermo Fisher Scientific, Waltham, Massachusetts, USA). About 50–75 ng (400-bp amplicons) and 250–300 ng (2 and 2.5-kb amplicons) of each SARS-CoV-2 isolate were treated with NEBNext Ultra II End repair / dA-tailing Module (New England Biolabs, Ipswich, Massachusetts, USA). The samples were then barcoded using EXP-NBD104 (barcodes 1–12) or EXP-NBD114 (barcodes 13–24) kits (Oxford Nanopore Technologies, Oxford, UK) and NEBNext Ultra II Ligation Master Mix (New England Biolabs, Ipswich, Massachusetts, USA). Barcoded samples were pooled and purified using 0.6 volume of AMPure XP magnetic beads. The AMII sequencing adapter (Oxford Nanopore Technologies, Oxford, UK) was ligated to about 75 ng (400-bp amplicons) or 300 ng (2 and 2.5-kb amplicons) of barcoded pools using Quick T4 DNA ligase (New England Biolabs, Ipswich, Massachusetts, USA) and the sequencing libraries were purified using 0.6 volume of AMPure XP magnetic beads. About 20 ng (400-bp amplicons) and 90 ng (2 and 2.5-kb amplicons) of the libraries were loaded on an R9.4.1 flow cell (FLO-MIN106). The sequencing was performed using a MinION Mk-1b device (Oxford Nanopore Technologies, Oxford, UK). For sequencing on the Flongle flow cells (FLO-FLG001), the library preparation was the same, except that one third to one half of the library was loaded compared to the amount used for the standard flow cell.

## Data processing

Nanopore sequencing data were base called and demultiplexed using Guppy v.3.4.4. Variant analysis was performed using Artic analysis pipeline v.1.1.3. (https://github.com/artic-network/artic-ncov2019) using recommended settings. The only exceptions were the minimum and maximum read lengths in the Artic guppyplex filter, which were set to 350 and 619 for the 400-bp amplicons and 1500 and 3000 for both the 2 and 2.5-kb amplicons, respectively. The goal of length filtering is to eliminate chimeric reads and short fragments, and thus the minimum and maximum are adapted to the expected amplicon lengths in the primer set used. We have used a more permissive setting for longer amplicons, as length deviations may possibly scale with amplicon length. Note that according to **Fig 2** reads failing due to length are relatively rare, particularly for 400-bp amplicons, and thus it does not seem that they were disadvantaged by stricter length filtering. For batch UKBA-2, the final sequences were produced by first combining sequencing reads from both standard and Flongle runs with the same primer set and running the Artic pipeline. Subsequently the results for the two primer sets were combined so that regions sufficiently covered by at least one amplicon set were

considered as finished. The same process was used in batch UKBA-3, but there was only data from standard flow cells available. Subsequent batches were based on 2 or 2.5-kb amplicons sequenced on a standard flow cell.

To compare different primer sets and flow cells, reads were also demultiplexed at the less strict default Guppy settings and aligned to various reference genomes by minimap2 v. 2.13-r852-dirty [41]. Reference genomes include the SARS-CoV-2 genome MN908947.3 [1], the human genome version hg19 downloaded from the UCSC genome browser [40], and the database for bacterial species typing included in the Japsa software [42]. To detect sub-genomic RNAs, reads were aligned to transcripts downloaded from the UCSC genome browser by minimap2, and classified as sub-genomic, if the alignment to a sub-genomic RNA has at least 5 matches more than the best alignment to the reference genome. An alignment to a sub-genomic RNA scores higher than an alignment to a genome if it spans the junction between the leader and the ORF portion of the RNA, as this junction does not occur in the genome. For purposes of visualization (**Fig 3**), randomly sampled reads classified as sub-genomic were aligned to the genome by BLAT [43]. Read coverage was computed using genomecov tool from BEDTools [44] with options -bga -split.

To compare the results for various sequencing data volumes, reads were ordered by the sequencing finish time and the initial portion with the desired total length was selected and used for the analysis in the Artic pipeline. To compare batches with a different number of samples, the cutoffs were expressed as the average amount per barcode.

## Ethics statement

The study has been approved by the Ethics committee of Biomedical Research Center of the Slovak Academy of Sciences, Bratislava, Slovakia (Ethics committee statement No. EK/BmV-02/2020). For all clinical specimens specifically collected for the purpose of this study, written informed consent has been obtained from the participants, and the appropriate institutional forms have been archived. In line with the statement of the Ethics committee, the consent was waived for samples previously collected for the purpose of primary diagnosis of SARS-CoV-2; these samples were made unidentifiable for the researchers performing this study.

## Supporting information

**S1 Table. Overview of the SARS-CoV-2 samples sequenced in this study.**
(PDF)

**S1 Fig. Dependence of the amount of missing sequence after Artic analysis on various sample properties.** (A) Cq value of the diagnostic RT-qPCR test, (B) DNA concentration after amplification, (C) length of storage of the sample before PCR amplification. Each dot corresponds to one sample, each sub-plot has a different level of sequencing per barcode.
(PDF)

**S2 Fig. Coverage along the genome in several MinION runs.** In all runs, an initial portion of the run containing on average 40-Mbp of sequencing data per barcode was used. Coverage values higher than 1000 were clipped at this value and are shown in blue. Coverage below 20 (default Artic cutoff) is shown in red. Medians of 10-bp windows are shown for smoothing.
(PDF)

**S3 Fig. Coverage along the genome for samples sequenced in other laboratories.** Data by the COVID-19 Genomics UK Consortium were downloaded from ENA archive project PRJEB37886 (https://www.ebi.ac.uk/ena/browser/view/PRJEB37886) on August 4, 2021. Two centers within this project, namely the University of Exeter and the University of Cambridge,

submitted a large number of samples amplified with 400-bp primer sets and sequenced by MinION sequencer (828 and 231 samples, respectively). Samples were grouped by submission dates and we randomly selected ten samples from submission dates with a large number of samples. We have sampled 20-Mbp of reads from each sample and aligned them to the reference. The plots show the coverage along the genome as in **Fig 4** and **S2 Fig**. Only 15-Mbp were used for sample ERR4671239 as more data was not available. Note that the downloaded reads are already filtered by barcode, size and are all alignable to the reference. In our 400-bp samples shown in **Fig 4** and **S2 Fig** each barcode has a different amount of data aligned due to differences in the quality of individual samples in the run, but the median is 18-Mbp, which is a value similar to the 20-Mbp cutoff used here.
(PDF)

## Acknowledgments

The authors wish to thank Lubomir Tomaska (Comenius University in Bratislava) for critical reading of the manuscript and discussions.

## Author Contributions

**Conceptualization:** Broňa Brejová, Tomáš Vinař, Jozef Nosek.

**Data curation:** Broňa Brejová, Askar Gafurov, Tomáš Vinař.

**Formal analysis:** Broňa Brejová, Askar Gafurov, Tomáš Vinař.

**Funding acquisition:** Broňa Brejová, Martina Neboháčová, Tomáš Vinař, Boris Klempa, Jozef Nosek.

**Investigation:** Kristína Boršová, Viktória Hodorová, Viktória Čabanová, Dominika Fričová.

**Methodology:** Kristína Boršová, Viktória Hodorová, Viktória Čabanová, Dominika Fričová, Jozef Nosek.

**Project administration:** Martina Neboháčová, Boris Klempa.

**Resources:** Boris Klempa.

**Writing – original draft:** Broňa Brejová, Tomáš Vinař, Jozef Nosek.

**Writing – review & editing:** Broňa Brejová, Kristína Boršová, Viktória Hodorová, Viktória Čabanová, Askar Gafurov, Dominika Fričová, Martina Neboháčová, Tomáš Vinař, Boris Klempa, Jozef Nosek.

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
