## [Decision Letter · Decision Letter 0]

30 May 2021

PONE-D-21-13987

Nanopore Sequencing of SARS-CoV-2: Comparison of Short and Long PCR-tiling Amplicon Protocols

PLOS ONE

Dear Dr. Nosek,

Thank you for submitting your manuscript to PLOS ONE. After careful consideration, we feel that it has merit but does not fully meet PLOS ONE’s publication criteria as it currently stands. Therefore, we invite you to submit a revised version of the manuscript that addresses the points raised during the review process.

We look forward to receiving your revised manuscript.

Kind regards,

Ronald Dijkman, PhD

Academic Editor

PLOS ONE

Journal Requirements:

Reviewers' comments:

Reviewer's Responses to Questions

**Comments to the Author**

1. Is the manuscript technically sound, and do the data support the conclusions?

Reviewer #1: Yes

Reviewer #2: Partly

2. Has the statistical analysis been performed appropriately and rigorously? 

Reviewer #1: Yes

Reviewer #2: N/A

3. Have the authors made all data underlying the findings in their manuscript fully available?

Reviewer #1: Yes

Reviewer #2: Yes

4. Is the manuscript presented in an intelligible fashion and written in standard English?

Reviewer #1: Yes

Reviewer #2: Yes

5. Review Comments to the Author

Reviewer #1: The study by Brejova and colleagues compared three versions of the PCR-tiling protocol using amplicon sizes of 400, 2000 and 2500 bp for sequencing SARSCoV-2 genomes from clinical samples on the nanopore MinION platform. Based on clinical samples obtained from 152 SARS-CoV-2

isolates from Slovakia obtained March 2020 to March 2021, the authors concluded that the protocol based on short 400-bp amplicons generally produces more usable data, but more variable coverage of the genomes. In comparison, it was easier to obtain close-to-finished genomes with longer amplicons, generally requiring smaller amounts of raw data produced per barcode sequenced. However, they also observed that protocols based on long amplicons produced a higher percentage of reads that are unsuitable for further analysis with the Artic pipeline.

The manuscript is clear and well written.

Major comments

A) Experimental design. In a comparative experiment, it would be recommended to compare the various methods on the exact same material. In the current study, the comparison was done while new samples were being accumulated. Therefore it is difficult to know how much of the observed differences may also be attributed to variation in the biological material itself or other sampling artefacts. In addition, did the authors test the sensitivity of the different protocols on the serial dilutions of the exact, same RNA material? This would also enable to see how the protocols perform on standardized material.

B) The Results section should comment more on the diversity of the lineages, and whether some batches were more diverse in terms of sequences as compared to that of the original Wuhan genome sequence. Would more diversity be correlated with more difficulty to assemble the genome sequences of the isolates?

C) Table S1. There are some reports of lineages "B.1", and "B1.1". Please check again those, as these are not officially attributed lineages, mostly due to poor classification.

D) the authors ordered the primer pool for short amplicons at IDT, but it is not clear whether the lower efficiency of the very high number of primers present in the 400-bp pool could be compensated by adjusting some of the primer concentrations. Although this is not practical for routine diagnostics, this would be essential to know for the current study because it is not clear whether the variability in coverage is due to issues with the primer competition within the pools the authors ordered (so study-specific) or with the 400-bp pools of the ARTIC v3 in general (generalizable to all studies using the ARTIC v3 protocol). There have been several communications (Itokawa et al. 2020 PLoS ONE, Gohl et al. 2020 BMC Genomics) over the last year about the need to readjust the pools when using 400-bp amplicons in case some regions are poorly covered. If you do so (and we did so too), you obtain a perfect coverage even with 400-bp amplicons.

E) The authors washed some of the flow cells with the nuclease buffer from ONT. Did they also check for carry-over between different runs? Could this also explain the "contaminants" observed on Page 9?

F) Page 9. While referring to possible contaminants in the observed data, did the authors also sequence non-template controls to detect potential contaminants from the buffers or reagents they used?

G) Figure 2. The legend describes some subpanels (A-G), but the figure shows only 2 panels (A-B).

H) Page 17. Data processing. Why did the authors do not use the same window size for filtering the amplicons by size: "Minimum and maximum read lengths in the Artic guppyplex filter were set to 350 and 619 for the 400-bp amplicons and 1500 and 3000 for both the 2 and 2.5-kb amplicons, respectively"? Did this have an impact on the selection of more data for the longer amplicons for instance?

I) Page 17. The analysis of subgenomic RNA is not convincing to me. What could be more convincing would be to show that the starts of the read alignments match the sub-genomic RNA start and not any genomic region of the reference for instance.

Minor comments

-The resolution of the figures was not optimal on the version I reviewed, which rendered the evaluation of the figures difficult.

-Introduction. I suggest reorganising the bottom of the first page as follows. Put "The virus sequences were determined using a range of experimental approaches based on metagenomics…modified nucleotides in the viral genome (5-7)." after the part

"Rapid, cost-effective, and near …still represents a challenging task (12-22)", so that when you introduce sequencing, it logically follows up with the paragraph about nanopore.

- last sentence of the introduction. The number of pores you indicate are the theoretical numbers at production, but not at delivery of the products.

-Figure 2. The vertical order of the samples in the figure does not follow the order of the rows in Table 1, so it is difficult to match the information from the table and figure.

- Table 1. The alignments between the first 2 columns are not clearly indicated, so it is not easy to clearly match the batch names and amplicon sizes at a first glance.

Reviewer #2: Brejová et al. have submitted a research article on the comparison of short and long PCR-tiling amplicon protocols targeted at SARS-CoV-2 using nanopore sequencing. Data generated using such protocols is used to identify variants of concern and, when made accessible to the scientific community and the public by submission to public databases, enables further analyses. Using combinations of three different primer sets and two flow-cell types, various sequencing results are presented.

While the article is pointing at interesting issues using the PCR-tiling amplicon protocols presented, there is no clear distinction made between the two long amplicon schemes. This article would benefit from a evaluation / recommendation on which scheme to use or an explanation on when to use which (long) amplicon scheme. Additional data may prove useful, especially on the 2.5kb protocol which was modified.

Page 2

Did you encounter issues with these in consensus generating due to human or bacterial contamination or was this only an issue in sequencing efficiency?

Page 3

Replace “currently” with a fixed timepoint.

Page 4

A 60-fold coverage is referenced to make Nanopore results comparable to Illumina results, while the default artic value, used in this study, is 20. Please elaborate your decision. “Pandemics” should be singular as the referenced study aims at SARS-CoV-2.

Remove “the” in “the rigorous comparison” and “the highly accurate”.

Use present tense throughout the last section.

Page 5

According to Table 1, no FLO-MIN106D was used. Please correct either the Table or the bracket after “MinION runs with the standard”.

Page 6

Replace “In some experiments” by specifying how many experiments and/or samples.

Use “and/or flow cells” instead of “and sequencing devices” as UKBA-4 compared only flow cells. Replace all "sequencing device" with "flow cells" as the sequencing device was always a MinION Mk1-b.

Page 7

Use “buffer containing nuclease”.

Table 1: indicate that “Flow-cell QC at start” is the amount of active pores.

Fig 1 The Figure legend (a) and (b) and the plot Y-title do not correspond. Replace “Flongle” and “Standard” by flow cell types (see Table 1) and use “run 1” or “run 2”, if necessary.

Page 8

S1 Figure: Specify unit used for amplicon concentration.

Rephrase the sentence beginning with “Majority of failed reads” to make clear that groups A-C contain incomplete *barcoding* and low quality. Incompleteness can be confused with short reads.

Use “seem to be more prevalent” instead of “are apparently more” as you are stating that there are no clear differences.

According to Fig 2A, up to 27% of the reads are non-target and not “up to 6% of reads”. Please check.

Page 9

Please rename the Figure 2 Title, as the Figure also shows fractions of reads which are not discarded. Use “D: reads that do not align” instead of “D: reads do not align”, the same for E and add “reads” to F. The possible explanations in brackets should go to Results and Discussion.

Figure 2B: The total read count does not seem informative in this context. Consider removing it.

Page 10

Table 2 Title Sub-genomic RNA fractions (in %): Fraction of what?

There is no (A) or (B) visible in the Table. If (A) is for the first table, then the legend is wrong as there are UKBA-2/-3/-4 and not UKBA-2, 2-kb amplicons.

Page 12

Fig 5 – Please state the N of samples used in the titles of the plots.

Fig 5 – Legend: 10% percentile should be 10th percentile. It is unclear to me what is meant written in brackets. Is there always the same 3-kb portion of the genome with the lowest coverage?

Replacement of the rightmost primer in the 2.5-kb scheme: did you find mismatches for the original primer?

For Fig 5 you state that the modification of 1 primer may result in fewer regions with insufficient coverage. Do you have data comparing the two protocols on the same samples to back this assumption? The data shown in Fig 5 for the 2.5kb schemes are two different batches.

Page 13

Reference to Fig 2A for the reads passing all filters and the discarded reads.

Fig 6, state in the box in the upper right, that for 2.5kb N=1 each.

Fig 6 legend: Did you replace the last primer pair or only the last primer? I would expect a gap when replacing the last primer pair.

Page 14

Consider rephrasing the sentence about the MinION platform and rather aim at the PCR-tiling amplicon approach.

Materials and Methods

Use consistent citations of producers/suppliers, fully name them at least when first mentioning them.

Data processing: replace the SOP link to artic with a github link as you reference to the tool itself. The recommended settings from the SOP are 400 to 700, while you used 350 and 619 and stated you used the recommended settings. Please adapt and/or explain.

The figures are quite blurry, which makes them difficult to read. Please make sure that they are well-readable.

6. PLOS authors have the option to publish the peer review history of their article (what does this mean?). If published, this will include your full peer review and any attached files.

Reviewer #1: **Yes: **Alban Ramette

Reviewer #2: No

---

## [Author Response · Author response to Decision Letter 0]

29 Jun 2021

Journal Requirements: 

Please ensure that your manuscript meets PLOS ONE's style requirements, including those for file naming. The PLOS ONE style templates can be found at https://journals.plos.org/plosone/s/file?id=wjVg/PLOSOne_formatting_sample_main_body.pdf and https://journals.plos.org/plosone/s/file?id=ba62/PLOSOne_formatting_sample_title_authors_affiliations.pdf

We modified the revised version according to the guidelines.

We provide more detailed information in the ethics statement.

As requested, the ethics statement was transferred to the Methods section.

Responses to Reviewers:

Reviewer #1: The study by Brejova and colleagues compared three versions of the PCR-tiling protocol using amplicon sizes of 400, 2000 and 2500 bp for sequencing SARSCoV-2 genomes from clinical samples on the nanopore MinION platform. Based on clinical samples obtained from 152 SARS-CoV-2

isolates from Slovakia obtained March 2020 to March 2021, the authors concluded that the protocol based on short 400-bp amplicons generally produces more usable data, but more variable coverage of the genomes. In comparison, it was easier to obtain close-to-finished genomes with longer amplicons, generally requiring smaller amounts of raw data produced per barcode sequenced. However, they also observed that protocols based on long amplicons produced a higher percentage of reads that are unsuitable for further analysis with the Artic pipeline.

The manuscript is clear and well written.

Major comments

A) Experimental design. In a comparative experiment, it would be recommended to compare the various methods on the exact same material. In the current study, the comparison was done while new samples were being accumulated. Therefore it is difficult to know how much of the observed differences may also be attributed to variation in the biological material itself or other sampling artefacts. In addition, did the authors test the sensitivity of the different protocols on the serial dilutions of the exact, same RNA material? This would also enable to see how the protocols perform on standardized material.

We agree. Therefore, in our initial experiments (i.e. the batches UKBA-2 and UKBA-3, Table 1), we compared the sequencing of both short (400-bp) and long (2-kb) amplicons originating from the same biological material. Moreover, in the batches UKBA-2 and UKBA-4, we tested the same sequencing libraries on different types of flow-cells (i.e. FLO-MIN106 and FLO-FLG001). We have clarified these facts in the results section.

On the other hand, we did not test the sensitivity of PCR-tiling protocols on serial dilutions of the standardized sample as this has been done in several recent studies. Rather, we preferred to test the sequencing protocols using real-world clinical samples differing by Cq values, storage, SARS-CoV-2 genotype, content of contaminating DNA, etc.

B) The Results section should comment more on the diversity of the lineages, and whether some batches were more diverse in terms of sequences as compared to that of the original Wuhan genome sequence. Would more diversity be correlated with more difficulty to assemble the genome sequences of the isolates?

The divergence of individual samples from the reference sequence ranges between approximately 0.02% to 0.13%, with higher divergence in case of newer samples. We did not observe this to have an effect on bioinformatics tools used in the analysis which are designed to work consistently at a much broader scale of divergence. We have added a discussion of this point to the Conclusion section.

C) Table S1. There are some reports of lineages "B.1", and "B1.1". Please check again those, as these are not officially attributed lineages, mostly due to poor classification. 

Pangolin classifications originally reported in the table were computed at the time of submission to GISAID. However, as Pangolin classification evolves, some of the samples are reassigned to newly created lineages. We have recomputed this column using Pangolin version from June 5, 2021, which changed the classification of eight samples. Some of the older samples are still classified as B.1.1, which is expected since many sublineages of B.1.1 did not exist at the time of collections of these samples. 

D) the authors ordered the primer pool for short amplicons at IDT, but it is not clear whether the lower efficiency of the very high number of primers present in the 400-bp pool could be compensated by adjusting some of the primer concentrations. Although this is not practical for routine diagnostics, this would be essential to know for the current study because it is not clear whether the variability in coverage is due to issues with the primer competition within the pools the authors ordered (so study-specific) or with the 400-bp pools of the ARTIC v3 in general (generalizable to all studies using the ARTIC v3 protocol). There have been several communications (Itokawa et al. 2020 PLoS ONE, Gohl et al. 2020 BMC Genomics) over the last year about the need to readjust the pools when using 400-bp amplicons in case some regions are poorly covered. If you do so (and we did so too), you obtain a perfect coverage even with 400-bp amplicons.

Naturally, as it has been reported before, an optimisation of the 400-bp primer scheme (e.g. by changing the primer positions or concentrations) may solve some of the issues associated with the uneven coverage. Yet, the V3 scheme comprises 218 primers and therefore for routine analysis outside of the large sequencing facilities it is impractical to synthesize, pool, test and readjust this complex primer pool. Rather, we consider it more convenient to work with the primer schemes for long amplicons containing only 34 (2-kb) or 28 (2.5-kb) primers. Discussion of this issue was added to the Conclusion section.

E) The authors washed some of the flow cells with the nuclease buffer from ONT. Did they also check for carry-over between different runs? Could this also explain the "contaminants" observed on Page 9?

In general, we noticed only negligible carry-over in the sequencing runs using nuclease-washed flow-cells. This conclusion is based on the comparison of data obtained from the washed and new FLO-MIN106 or FLO-FLG001 in the batches UKBA-2, UKBA-3 and UKBA-4. Moreover, in experiments with washed flow-cells (e.g. UKBA-2, UKBA-3), samples unrelated to SARS-CoV-2 were sequenced (i.e. yeast DNA) or different barcode sets (NB01-NB12 vs. NB13-NB24) were used. 

As far as the “contaminants” are concerned, these are unrelated to any sample processed in the sequencing lab and clearly appear to be associated with human samples (e.g. human DNA or DNA from bacteria associated with the respiratory tract). Moreover, the flow cell used for the detailed analysis of contaminants (UKBA-2, 2-kb amplicons, Fig 2B) was new, not flushed.

F) Page 9. While referring to possible contaminants in the observed data, did the authors also sequence non-template controls to detect potential contaminants from the buffers or reagents they used?

Non-template controls were included only in the RT-qPCR assays. However, the same buffers/kits were used in multiple sequencing experiments and the “contaminants” were present only in those particular samples (even within a single batch processed simultaneously). As mentioned above, identified bacteria are commonly associated with the human respiratory tract pointing to their origin in the clinical samples (e.g. oropharyngeal swabs) used for the SARS-CoV-2 sequencing.

G) Figure 2. The legend describes some subpanels (A-G), but the figure shows only 2 panels (A-B).

The letters A-G did not refer to panels but to groups of reads. As the panels were labeled A and B creating ambiguity, we have changed read group labels to (a)-(g) and modified the legend accordingly. Thank you for pointing this out.

H) Page 17. Data processing. Why did the authors do not use the same window size for filtering the amplicons by size: "Minimum and maximum read lengths in the Artic guppyplex filter were set to 350 and 619 for the 400-bp amplicons and 1500 and 3000 for both the 2 and 2.5-kb amplicons, respectively"? Did this have an impact on the selection of more data for the longer amplicons for instance?

For 400-bp the size limits mostly follow Artic guidelines, with the maximum set to 200-bp above the longest amplicon, as recommended, and the minimum is slightly lower than the minimum amplicon length, to allow possible deletions (both sequencing errors and deletions on the sequenced sample). However, as these guidelines were developed for shorter amplicons, we have used a more permissive setting for longer amplicons, as length deviations may possibly scale with amplicon length. However, if we take all reads passing all filters and consider their lengths, most reads fit into a much narrower range. For example, after dropping 1% of shortest and 1% of longest reads, with 400-bp amplicons we get range 468-543-bp, with 2kb amplicons 1589-2169-bp, with original 2.5-kb amplicons 1860-2747-bp. Some of the shorter reads are due to sub-genomic RNAs. Also note that according to Fig.2 reads failing due to length are relatively rare, particularly for 400-bp amplicons, and thus it does not seem that they were disadvantaged by stricter length filtering. We have added a note to this effect to the article.

I) Page 17. The analysis of subgenomic RNA is not convincing to me. What could be more convincing would be to show that the starts of the read alignments match the sub- genomic RNA start and not any genomic region of the reference for instance.

The 5' end leader sequence of sub-genomic mRNAs is short, spanning only 45-bp within our amplicons. Due to sequencing errors, it is difficult to align it on its own to nanopore reads, particularly with fast alignment tools, such as minimap2, which are tailored to searching for long alignments. This is the reason why we align the reads to both the full genome and sub-genomic RNAs and classify a read as sub-genomic, if it aligns to the sub-genomic RNA better than to the genome. An alignment to a subgenomic RNA scores higher than an alignment to a genome if it spans the junction between the leader and the ORF portion of the RNA, as this junction does not occur in the genome. This justification was added to the Methods section. To illustrate this further, we have taken a small random sample of 2-kb reads, which were classified as sub-genomic and aligned them to the genome by the BLAT tool, which attempts to do spliced alignments. The result is shown in a new version of Figure 3. BLAT sometimes creates spurious short exons; but aside from that the alignments nicely follow the correct location of subgenomic RNAs. 

Minor comments

-The resolution of the figures was not optimal on the version I reviewed, which rendered the evaluation of the figures difficult.

The lower resolution of figures is caused by the journal submission system and is beyond our control. We submitted high-resolution figures in the .tiff format which are accessible from the hyperlinks present above each figure. We believe that the camera-ready produced by the journal will not suffer from these problems.

-Introduction. I suggest reorganising the bottom of the first page as follows. Put "The virus sequences were determined using a range of experimental approaches based on metagenomics...modified nucleotides in the viral genome (5-7)." after the part "Rapid, cost-effective, and near ...still represents a challenging task (12-22)", so that when you introduce sequencing, it logically follows up with the paragraph about nanopore.

Modified as suggested and the corresponding references were reordered.

- last sentence of the introduction. The number of pores you indicate are the theoretical numbers at production, but not at delivery of the products.

Yes, these are “nominal” numbers of pores in the flow-cells (FLO-MIN106 and FLO-FLG001). The actual number of active pores in each flow-cell used in our experiments are shown in Table 1 (column denoted ”Flow-cell QC”). 

-Figure 2. The vertical order of the samples in the figure does not follow the order of the rows in Table 1, so it is difficult to match the information from the table and figure. - Table 1. The alignments between the first 2 columns are not clearly indicated, so it is not easy to clearly match the batch names and amplicon sizes at a first glance.

The order of samples in both Figure 1 and Figure 2 were changed according to Table 1. To facilitate distinguishing among individual UKBA batches in Table 1 we added horizontal lines between them.

Reviewer #2: Brejová et al. have submitted a research article on the comparison of short and long PCR-tiling amplicon protocols targeted at SARS-CoV-2 using nanopore sequencing. Data generated using such protocols is used to identify variants of concern and, when made accessible to the scientific community and the public by submission to public databases, enables further analyses. Using combinations of three different primer sets and two flow-cell types, various sequencing results are presented.

While the article is pointing at interesting issues using the PCR-tiling amplicon protocols presented, there is no clear distinction made between the two long amplicon schemes. This article would benefit from a evaluation / recommendation on which scheme to use or an explanation on when to use which (long) amplicon scheme. Additional data may prove useful, especially on the 2.5kb protocol which was modified.

Our main focus was the comparison of 400-bp and 2-kb amplicon pools. Unfortunately, we do not have additional data directly comparing 2-kb and 2.5-kb protocols. At present, we use modified 2.5-kb protocol in our genomic surveillance pipeline. We now mention this fact in the conclusion.

Page 2

Did you encounter issues with these in consensus generating due to human or bacterial contamination or was this only an issue in sequencing efficiency?

Only reads aligning to the SARS-CoV-2 genome and having appropriate lengths were used for variant calling and subsequent consensus generation. Thus the problem with human or bacterial contamination is mainly decreased sequencing efficiency. We have reformulated the abstract to make this clearer. 

Page 3

Replace “currently” with a fixed timepoint.

Corrected as suggested and updated (as of June 22, 2021)

Page 4

A 60-fold coverage is referenced to make Nanopore results comparable to Illumina results, while the default artic value, used in this study, is 20. Please elaborate your decision. 

The coverage threshold 20 is implicitly hardcoded into artic minion command from the Artic pipeline, and it cannot be changed without modifying the source code. Nanopolish, which is the underlying variant caller, also uses coverage 20 as a default cutoff. We have not considered modifying this threshold from the default values. In our experience, the Artic pipeline is relatively conservative and masks some variants by ambiguous base N at lower coverages, but only rarely do we see incorrect predictions.

“Pandemics” should be singular as the referenced study aims at SARS-CoV-2.

Corrected.

Remove “the” in “the rigorous comparison” and “the highly accurate”.

Corrected.

Use present tense throughout the last section.

Corrected. 

Page 5

According to Table 1, no FLO-MIN106D was used. Please correct either the Table or the bracket after “MinION runs with the standard”.

Corrected.

Page 6

Replace “In some experiments” by specifying how many experiments and/or samples.

Corrected.

Use “and/or flow cells” instead of “and sequencing devices” as UKBA-4 compared only flow cells. 

Corrected.

Replace all "sequencing device" with "flow cells" as the sequencing device was always a MinION Mk1-b.

Corrected.

Page 7

Use “buffer containing nuclease”.

Corrected.

Table 1: indicate that “Flow-cell QC at start” is the amount of active pores.

Corrected.

Fig 1 The Figure legend (a) and (b) and the plot Y-title do not correspond. Replace “Flongle” and “Standard” by flow cell types (see Table 1) and use “run 1” or “run 2”, if necessary.

In the legend, the word "fraction" was replaced by "percentage" and the flow cell types were changed as suggested, in both Figure 1 and 2.

Page 8

S1 Figure: Specify unit used for amplicon concentration.

Corrected.

Rephrase the sentence beginning with “Majority of failed reads” to make clear that groups A-C contain incomplete *barcoding* and low quality. Incompleteness can be confused with short reads.

Corrected.

Use “seem to be more prevalent” instead of “are apparently more” as you are stating that there are no clear differences.

Corrected.

According to Fig 2A, up to 27% of the reads are non-target and not “up to 6% of reads”. Please check.

Thank you very much for pointing out this mistake. We have checked the data; the figure is correct, and the text was modified to 27%.

Page 9

Please rename the Figure 2 Title, as the Figure also shows fractions of reads which are not discarded. Use “D: reads that do not align” instead of “D: reads do not align”, the same for E and add “reads” to F. The possible explanations in brackets should go to Results and Discussion.

We have clarified in the legend that the plots show also reads not discarded. The names for individual groups were modified as suggested. 

Figure 2B: The total read count does not seem informative in this context. Consider removing it.

We have removed the read count.

Page 10

Table 2 Title Sub-genomic RNA fractions (in %): Fraction of what?

We have rearranged the legend to make this point clearer.

There is no (A) or (B) visible in the Table. If (A) is for the first table, then the legend is wrong as there are UKBA-2/-3/-4 and not UKBA-2, 2-kb amplicons.

Labels (A) and (B) were changed to Top and Bottom. The text of the legend was rearranged to hopefully decrease the chance of misunderstanding.

Page 12

Fig 5 – Please state the N of samples used in the titles of the plots.

The information was added.

Fig 5 – Legend: 10% percentile should be 10th percentile. It is unclear to me what is meant written in brackets. Is there always the same 3-kb portion of the genome with the lowest coverage?

The legend was changed to 10th percentile and the text in the parenthesis was reformulated. The 3-kb portion is not the same in all cases.

Replacement of the rightmost primer in the 2.5-kb scheme: did you find mismatches for the original primer?

We did not find any mismatches in the left primer; the position of this primer is covered by an amplicon in the other PCR pool. We are unable to say if there are any mismatches in the right primer, as this region was not sequenced other than from the primer. 

For Fig 5 you state that the modification of 1 primer may result in fewer regions with insufficient coverage. Do you have data comparing the two protocols on the same samples to back this assumption? The data shown in Fig 5 for the 2.5kb schemes are two different batches.

Unfortunately, at present we do not have data that would allow direct comparison of two versions of the 2.5-kb protocol.

Page 13

Reference to Fig 2A for the reads passing all filters and the discarded reads.

Reference to the figure was added.

Fig 6, state in the box in the upper right, that for 2.5kb N=1 each.

Information added.

Fig 6 legend: Did you replace the last primer pair or only the last primer? I would expect a gap when replacing the last primer pair.

We replaced both primers of the right-most amplicon. There is no gap between the penultimate (2.5-kb) and new (2-kb) right-most amplicons, although the latter one is shorter on the side corresponding to the 3’ end of the viral genome.

Page 14

Consider rephrasing the sentence about the MinION platform and rather aim at the PCR-tiling amplicon approach.

The sentence has been rephrased to aim at the combination of PCR-tiling and nanopore sequencing.

Materials and Methods

Use consistent citations of producers/suppliers, fully name them at least when first mentioning them.

Corrected as suggested.

Data processing: replace the SOP link to artic with a github link as you reference to the tool itself. 

Corrected as suggested.

The recommended settings from the SOP are 400 to 700, while you used 350 and 619 and stated you used the recommended settings. Please adapt and/or explain.

According to our understanding, lengths 400 and 700 are examples; the same lengths are used also in Ebola SOP https://artic.network/ebov/ebov-bioinformatics-sop.html Both Ebola and NCoV further state "Try the minimum lengths of the amplicons as the minimum, and the maximum length of the amplicons plus 200 as the maximum." Amplicon lengths of V3 400bp Arctic protocol range from 380 to 419, so the upper limit of 619 was selected exactly based on the recommendations. We have slightly lowered the lower limit (from 380 to 350) to accommodate deletions (both sequencing errors and real deletions in the sample). It may also allow the use of some sub-genomic RNAs. For 2-kb and 2.5-kb amplicons, we have used a wider range of lengths; nonetheless this range is sufficient to filter out various short fragments as well as chimeric reads consisting of two full amplicons. We have made it clearer in the methods section that the length settings are not default. 

The figures are quite blurry, which makes them difficult to read. Please make sure that they are well-readable.

The lower resolution of figures is caused by the journal submission system and is beyond our control. We submitted high-resolution figures in the .tiff format and these were accessible from the hyperlinks present above each figure. We believe that the camera-ready version produced by the journal will not suffer from these problems.

---

## [Decision Letter · Decision Letter 1]

14 Jul 2021

PONE-D-21-13987R1

Nanopore sequencing of SARS-CoV-2: Comparison of short and long PCR-tiling amplicon protocols

PLOS ONE

Dear Dr. Nosek,

Thank you for submitting your manuscript to PLOS ONE. After careful consideration, we feel that it has merit but does not fully meet PLOS ONE’s publication criteria as it currently stands. Therefore, we invite you to submit a revised version of the manuscript that addresses the points raised during the review process.

We look forward to receiving your revised manuscript.

Kind regards,

Ronald Dijkman, PhD

Academic Editor

PLOS ONE

Journal Requirements:

Reviewers' comments:

Reviewer's Responses to Questions

**Comments to the Author**

1. If the authors have adequately addressed your comments raised in a previous round of review and you feel that this manuscript is now acceptable for publication, you may indicate that here to bypass the “Comments to the Author” section, enter your conflict of interest statement in the “Confidential to Editor” section, and submit your "Accept" recommendation.

Reviewer #1: All comments have been addressed

Reviewer #2: All comments have been addressed

Reviewer #3: (No Response)

2. Is the manuscript technically sound, and do the data support the conclusions?

Reviewer #1: Yes

Reviewer #2: Yes

Reviewer #3: Yes

3. Has the statistical analysis been performed appropriately and rigorously? 

Reviewer #1: Yes

Reviewer #2: N/A

Reviewer #3: N/A

4. Have the authors made all data underlying the findings in their manuscript fully available?

Reviewer #1: Yes

Reviewer #2: Yes

Reviewer #3: Yes

5. Is the manuscript presented in an intelligible fashion and written in standard English?

Reviewer #1: Yes

Reviewer #2: Yes

Reviewer #3: Yes

7. PLOS authors have the option to publish the peer review history of their article (what does this mean?). If published, this will include your full peer review and any attached files.

Reviewer #1: **Yes: **Alban Ramette

Reviewer #2: No

Reviewer #3: No

6. Review Comments to the Author

Reviewer #2: (No Response)

Reviewer #3: "Nanopore sequencing of SARS-CoV-2: Comparison of short and long PCR-tiling amplicon protocols" by Brejova et al. describes the comparison of different amplification methods for tiled-amplicon SARS-CoV-2 genome sequencing on the Oxford Nanopore MinION or Flongle platforms. The text is well-written and the data is clearly presented. I am somewhat puzzled by the poor performance of the ARTIC 400 bp amplicons in the authors' hands as this protocol has been used extensively and we routinely achieve much more uniform and complete coverage using this protocol (both on Illumina and ONT platforms). Some comparison to the coverage reported here using standard ARTIC primers and published data from other groups should be provided (possibly in the discussion section), to help the reader understand whether this is expected performance of the ARTIC primers or if their data is an outlier for some reason. Despite this, the authors demonstrate a clear advantage in using longer amplicons for Nanopore sequencing (at least in their hands) and these advantages (along with other perks, such as being able to use fewer primers and making re-balancing of pools easier) are clearly articulated in the manuscript. The authors have done a good job of responding to the previous reviewers' comments and I support publication of this manuscript if this additional point below can be addressed.

Major:

-As stated above, the authors should include a discussion of how their ARTIC 400 bp results compare to the balance and coverage obtained by other groups.

Minor:

-Figure 5. Showing the percent genome coverage (at some threshold like 10x or 20x) would be a more informative metric here as one could still have high average coverage, but low total genome coverage if the reads are skewed to a subset of the amplicons. However, I think it is fine if these plots remain unchanged as the green dots give a sense of what coverage looks like for the less well-represented amplicons.

---

## [Author Response · Author response to Decision Letter 1]

6 Aug 2021

Reviewer #3: "Nanopore sequencing of SARS-CoV-2: Comparison of short and long PCR-tiling amplicon protocols" by Brejova et al. describes the comparison of different

amplification methods for tiled-amplicon SARS-CoV-2 genome sequencing on the Oxford Nanopore MinION or Flongle platforms. The text is well-written and the data is clearly

presented. I am somewhat puzzled by the poor performance of the ARTIC 400 bp amplicons in the authors' hands as this protocol has been used extensively and we routinely achieve much more uniform and complete coverage using this protocol (both on Illumina and ONT platforms). Some comparison to the coverage reported here using standard ARTIC primers and published data from other groups should be provided (possibly in the discussion section), to help the reader understand whether this is expected

performance of the ARTIC primers or if their data is an outlier for some reason. Despite this, the authors demonstrate a clear advantage in using longer amplicons for

Nanopore sequencing (at least in their hands) and these advantages (along with other perks, such as being able to use fewer primers and making re-balancing of pools easier)

are clearly articulated in the manuscript. The authors have done a good job of responding to the previous reviewers' comments and I support publication of this manuscript if

this additional point below can be addressed.

Major:

-As stated above, the authors should include a discussion of how their ARTIC 400 bp results compare to the balance and coverage obtained by other groups.

Uneven coverage with ARTIC 400-bp protocol has already been observed by others in literature (e.g. [28, 31]), so this pattern is not specific to our experiments. We have also downloaded several data sets from COG UK member groups and confirmed that similar patterns occur in their data as well (now added as S3 Fig). We now summarize this in the conclusion section of the revised manuscript.

Minor:

-Figure 5. Showing the percent genome coverage (at some threshold like 10x or 20x) would be a more informative metric here as one could still have high average coverage,

but low total genome coverage if the reads are skewed to a subset of the amplicons. However, I think it is fine if these plots remain unchanged as the green dots give a sense

of what coverage looks like for the less well-represented amplicons.

The proportion of the genome with coverage below threshold 20x can be seen in coverage plots shown in Fig 4 and S2 Fig (red lines). We have referenced both figures in the caption of Fig 5 to make this connection more apparent.

---

## [Decision Letter · Decision Letter 2]

18 Oct 2021

Nanopore sequencing of SARS-CoV-2: Comparison of short and long PCR-tiling amplicon protocols

PONE-D-21-13987R2

Dear Dr. Nosek,

We’re pleased to inform you that your manuscript has been judged scientifically suitable for publication and will be formally accepted for publication once it meets all outstanding technical requirements.

Kind regards,

A. M. Abd El-Aty

Academic Editor

PLOS ONE

Additional Editor Comments (optional):

Reviewers' comments:

Reviewer's Responses to Questions

**Comments to the Author**

1. If the authors have adequately addressed your comments raised in a previous round of review and you feel that this manuscript is now acceptable for publication, you may indicate that here to bypass the “Comments to the Author” section, enter your conflict of interest statement in the “Confidential to Editor” section, and submit your "Accept" recommendation.

Reviewer #3: All comments have been addressed

2. Is the manuscript technically sound, and do the data support the conclusions?

Reviewer #3: Yes

3. Has the statistical analysis been performed appropriately and rigorously? 

Reviewer #3: N/A

4. Have the authors made all data underlying the findings in their manuscript fully available?

Reviewer #3: Yes

5. Is the manuscript presented in an intelligible fashion and written in standard English?

Reviewer #3: Yes

6. Review Comments to the Author

Reviewer #3: All of my previous comments have been satisfactorily addressed by the authors in the revised manuscript.

7. PLOS authors have the option to publish the peer review history of their article (what does this mean?). If published, this will include your full peer review and any attached files.

Reviewer #3: No

---

## [Editor Report · Acceptance letter]

21 Oct 2021

PONE-D-21-13987R2 

Nanopore sequencing of SARS-CoV-2: Comparison of short and long PCR-tiling amplicon protocols 

Dear Dr. Nosek:

I'm pleased to inform you that your manuscript has been deemed suitable for publication in PLOS ONE. Congratulations! Your manuscript is now with our production department. 

Kind regards, 

on behalf of

Prof. A. M. Abd El-Aty 

Academic Editor

PLOS ONE